# TCQG—Software-Defined Transmission Control Scheme in 5G Networks from Queuing Game Perspective

**DOI:** 10.3390/s19194170

**Published:** 2019-09-26

**Authors:** Chao Guo, Cheng Gong, Juan Guo, Haitao Xu, Long Zhang

**Affiliations:** 1Electronic and Communication Engineering, Beijing Electronics Science and Technology Institute, Beijing 100070, China; 2School of Computer and Communication Engineering, University of Science and Technology Beijing, Beijing 100083, China; cgong1986@gmail.com (C.G.); xuhaitao@ustb.edu.cn (H.X.); 3Guilin University of Electronic Technology-Vocational and Technical College, Beihai 536000, China; guojuan10802@163.com; 4School of Information and Electrical Engineering, Hebei University of Engineering, Handan 056038, China; longzhang011@gmail.com

**Keywords:** 5G, transmission control, software-defined, queuing game theory

## Abstract

The efficient processing and forwarding of big data is one of the key problems and challenges facing the next generation wireless communication network. Using a software definition method to virtualize the network can improve the efficiency of network operation and reduce the cost of network operation and maintenance. A software-defined transmission control scheme was presented to solve the excessive controller flow problem for 5G networks. Based on the queuing game theory, a system model was built due to the competition among the requests of the switch. The transmission control platform was in charge of resource allocation. It got maximum social welfare under a profit-maximizing fee. In this model, the optimal queue length was calculated and discussed in a first-come-first-served and last-come-first-served with preemption discipline. The optimal queue length was obtained and the optimal admission fee was calculated. Then, the single switch single controller transmission control model was extended to the multi-switches single controller model. As a result, the social welfare of the system containing the controller’s profit and switch surplus reaches the maximum.

## 1. Introduction

The characteristics of next-generation (5G) wireless networks are super high rate, ubiquitous network, low power consumption and low delay [1,2]. On the basis of the 4G network, the network data scale increases dramatically for the extension of the nature of connecting everything. At the same time, 5G/5G+ wireless network scenarios applications such as driverless cars, drone base stations, balloons, mobile edge computing, virtual reality, face recognition, artificial intelligence and so on appear constantly. Due to the computation-intensive and latency-sensitive features of these applications, Network Function virtualization (NFV) is considered to improve the operation efficiency of the 5G network [3,4]. NFV achieves different network functions by creating multiple virtual machines on existing hardware devices, which can significantly reduce the cost of network hardware equipment, function maintenance and improve the efficiency of information transmission. Therefore, it has attracted wide attention from academia and industry. The need for efficient forwarding and processing of big data puts forward higher requirements for the transmission control mechanism of the network. The benefits gained from existing hardware can be maximized by the dynamic mapping of virtual resources onto the physical hardware such as Virtual Network Embedding (VNE) resource allocation algorithms [5,6,7]. In order to improve network efficiency, the software-defined network is proposed based on the concept of NFV.

In the software-defined network (SDN), the control logic of the network (control plane) is separated from the underlying routers and switches (data plane) that forward traffic [8,9]. The switch becomes a simple forwarding device, the control logic is implemented in the controller (or network operating system) and the policy implementation and network configuration and evolution are simplified. However, the load pressure of the controller node increases with the forwarding of a lot of control information. The primary goal of most applications is to be designed by minimizing power consumption, maximizing total network utilization, providing optimal load balancing and other generic resource optimization techniques [10]. These schemes consider the promotion of several network parameters in a single or limited combination and cannot conduct systematic modeling on the network comprehensively, so as not to obtain the optimal strategy [10,11,12]. In order to improve network performance, transmission control should be paying more attention to. Business requests on the switch join a queue of the controller node depending on their own information. The purpose of each switch is to achieve the maximum transmission of information under the premise of satisfying their demands. In the case of limited resources such as computing, storage, bandwidth, and others, competition arises between network nodes including the switch and the controller nodes in SDN. How to improve the efficiency of data forwarding by means of optimization strategy is the key problem that the 5G network technology needs to consider. Game theory has been studied and applied in the field of optimal allocation of network resources with competitive relations [13,14,15]. The behavior and objective of multi-player competition can be abstracted as a game model and then the equilibrium solution can be obtained by solving the model. The reasonable game model will provide an efficient foundation for engineering design.

The key problem of the transmission control mechanism is to improve the negative impact of network bottleneck bandwidth on the overall performance of the system [16,17,18]. In this paper, a system model of software-defined 5G networks is built by using the queuing game theory [19]. It is used to establish a mathematical model of the competitive behavior and target between SDN nodes. The optimal strategy of the nodes is obtained by solving the theoretical equilibrium solution of the model, so as to relieve the transmission pressure of controller nodes which are easy to be congested and realize the maximum network efficiency. Here, the behavior of nodes is divided into the following two categories: whether the request of switching nodes join to the queue of the controller node and the admission fee value set by the controller node. These two kinds of behaviors are related to each other. The goal of the system is to maximize data transmission within the network carrying capacity. The objective is abstracted as an effect function, which consists of the benefit obtained from completing the data service, the admission fee to be paid and the waiting cost. Such a design takes into account the throughput and delay of the important network indicators in SDN and constructs the optimal system strategy by game among nodes. At the same time, due to the independence of the model in switch request parameters design, it is conducive to system upgrading according to specific network requirements in the future, and more suitable for the future changeable individual requirements of network function virtualization. The main drawbacks of the solution are the bottleneck bandwidth and the limitation of end-to-end delay on network performance. At last, an optimal software-defined transmission control algorithm is proposed based on queuing game theory over the 5G network (TCQG) to improve the network performance. The system model is formulated based on the queuing game theory to get the optimal admission fee by maximizing the controller’s profit and social welfare of the TCPL. Taking the multi-switches single controller model into account, the variable arrival rates are introduced for modifying the system model. The main contributions can be summarized as follows:Queuing game theory is introduced to obtain the optimal access strategy in the controller queue and the complexity of the transmission control algorithm is reduced by transferring scheme design to initial theoretical calculation.The controller and switch are considered as equal game players to reduce the dependence on a certain role.The single switch single controller transmission control model was extended to the multi-switches single controller model.

The rest of this paper is briefly described below. Section 2 is an introduction to some related works. Section 3 studies the theoretical framework and the system model in which the switch request decides whether or not to join a queue of the controller. The resource optimization of the controller and the TCPL are formulated in this section also. Section 4 proposes the TCQG algorithm for the single switch single controller model and extends it to the multi-switch single controller model over SDN. Section 5 deals with simulations and comparisons. At last, a conclusion is drawn in Section 6.

## 2. Related Work

In order to improve the resource allocation efficiency of 5G wireless network virtualization, the SDN framework is considered [20]. SDN originated from the research project of Stanford University by Clean Slate in 2006 [21] and the concept was proposed in 2009 [22]. The focus is the resource optimization of 5G using software-defined conception, which is being continuously improved and the application of flow engineering is also expanding.

As for the negative impact of bottleneck bandwidth on network performance, some research is devoted to improving the transmission control mechanism of SDN. In Reference [23], the authors considered traffic Heterogeneity defined by the rate between numbers of UDP over TCP traffic flows for each forwarding device in ultra-dense 5G wireless networks. Novel three heuristics including shortest path and e2eDelay optimization algorithms running in a parallel manner was proposed to accelerate centralized SDN-Controller. In Reference [24], an SDN-based TCP congestion control mechanism (SDTCP) was designed by leveraging features such as centralized control methods and the global view of the network to solve the TCP congestion problems. Once the switch congestion occurs, the controllers could select the background flows and reduce their bandwidth by adjusting the advertised packets window of the corresponding background flows to reserve more bandwidth for burst flows. Both schemes mitigated the TCP congestion problem by differentiating between different business flows. However, the distinction between only two types is not enough to include a large number of flow types with different types and characteristics in the era of big data. These studies provide some inspirations for the system model of designing a universal transmission control mechanism. According to the benefit of service obtained by the requests, different flow types are distinguished and then the network transmission model is established through game theory to solve the resource allocation problem.

In the past few years, a number of studies have investigated the problem of load balancing in SDN. In Reference [25], an SDN-enhanced Intercloud Manager (S-ICM) that allocates network flows was introduced for a congested cloud network. It consisted of monitoring and decision making parts. For monitoring, S-ICM uses the SDN control message that observes and collects data and decision-making is based on the measured network delay of packets. It showed that S-ICM is better at avoiding system saturation than HFA and RR under heavy load formula using the RR job scheduler. In Reference [26], the authors used a software-defined networking controller with a global view to facilitating dynamic virtual resource allocation and content caching in the architecture of software-defined information-centric network virtualization with device-to-device (D2D) communications. Considering the inaccurate channel estimation and measurement, the virtual resource allocation and caching optimization were formulated as a discrete stochastic optimization problem including imperfect channel state information. It was effective under different system parameters with extensive simulations. Both schemes extract global network information from the controller and take it as a decision basis. However, the excessive dependence on the controller’s function may lead to a decrease in the robustness of the system. In this paper, the controller and switch are considered as equal game players to reduce the dependence on a certain role.

In addition, there has been some research in the literature [27,28,29] analyzing the performance of SND from different perspectives such as energy management, orchestrate bandwidth, and requests game. The primary goal of most schemes is minimizing power consumption, maximizing total network utilization and providing optimal load balancing. In Reference [27], an Energy Monitoring and Management Application (EMMA) was proposed to minimize the energy consumption of the backhaul network. It tried to shut down the idle nodes and concentrated traffic on the smallest possible set of links, which in turn increases the number of idle nodes. EMMA provided excellent energy-saving performance, especially in larger network scenarios. In Reference [28], an SDN-based resource allocation framework was proposed that can properly orchestrate heterogeneous radio bandwidth in emerging LTE/WLAN multi-radio networks in a centralized and holistic manner. State-of-the-art smartphones can utilize such heterogeneous LTE and WLAN radio bandwidth by operating the multi-radio interfaces simultaneously. Price-based heterogeneous resource allocation algorithms in LTE/WLAN multi-radio networks were also proposed and achieved network utilization and fairness through allocating heterogeneous radio resources to multi-radio smartphone users. In Reference [29], by utilizing mean-field game theory, the authors presented an original discrete-time, distributed, a non-cooperative load-balancing algorithm to balance the requests of the switches dynamically among the SDN controllers. A specific equilibrium among loads of the providers was proven to converge to an arbitrarily small neighborhood. In order to improve the scalability of SDN networks, SDN Proxies were also introduced by dynamically dispatching the control workload across the available SDN controllers. These schemes reduce network energy consumption and improve global efficiency by optimizing some parameters of the network. However, some design conditions are not conducive to the optimization of algorithm complexity. The basic unit of network transmission control is the processing mechanism and efficiency of the node queue. In the TCQG scheme, queuing game theory is introduced to obtain the optimal access strategy in the controller queue and the complexity of the transmission control algorithm is reduced by transferring scheme design to initial theoretical calculation.

In this paper, the optimal software-defined transmission control scheme is proposed by confirming the area of the controller’s queue length. The switch’s benefit, controller’s profit and social welfare of the transmission control platform (TCPL) are formulated. On this basis, the optimal queue length is calculated and discussed in first-come-first-served and last-come-first-served with preemption discipline. The optimal admission fee is obtained in relation to the queue length. It assumes that each switch request gets its own benefit after completing service. The controller announces the admission fee to the switch, the switch then calculates each request’s minimum benefit through subtracting the admission fee and the cost of queuing. If the request’s benefit is non-negative, the switch request joins the controller to be serviced. In this model, how to find the optimal admission fee is the key. It formulates the controller’s profit, maximizes the value and iteratively modifies under the first-come-first-served (FCFS) and last-come-first-served with preemption (LCFS-PR) discipline to get the unique queue length which is the threshold value. In the meanwhile, the social welfare of the TCPL is obtained with the queue length. According to the relationship between the threshold queue length and the admission fee, the optimal admission fee is obtained finally. With the above design, a novel software-defined transmission control scheme of the single switch single controller is modeled over the 5G network. Based on it, how to extend the result to the multi-switch single controller model is discussed. The most important distinction between the two models is the different arrival rates of the TCPL. Therefore, the improved model is built to solve the transmission control problem under the situation of a multi-switch single controller. At last, the effective software-defined transmission control algorithm is designed for promoting network performance in 5G networks.

## 3. System Model and Problem Description

The 5G network connects everything wirelessly and the diversity of hardware devices may limit its development. By separating the functions of hardware and software, the bottleneck of network upgrades can be reduced and efficient network transmission can be realized. The network model concept of SDN is used to virtualize 5G network functions. The system architecture includes the data layer and the control layer, in which the switch in the data layer is responsible for connecting various hardware devices and forwarding data and the controller in the control layer is responsible for managing the switch and strategically allocating resources.

### 3.1. System Model

To simplify the transmission control problem, a single switch single controller in SDN is assumed and then expend the result of the system model to the situation of a multi-switches single controller. A TCPL is built for resource allocation and calculation. While the switch requests come for service, the TCPL returns an admission fee. If the switch’s benefit is non-negative, it enters the queue of the controller. The buffer length of the controller is finite. Therefore, a suitable queue length needed to be suggested for the optimal system welfare.

A Transmission control system model using the TCQG algorithm is shown in Figure 1. The user terminals are connected to the network through 5G base station and the data plane switches are connected to the small base station network. Data information is transmitted between switches. Multiple switches are subordinate to an SDN controller, transmitting control information, obtaining rules or other control instructions for data flow. The TCPL consists of a multi-switches single controller and the TCQG algorithm is applied to the transmit link from the switch to the controller which is prone to congestion. Each controller has a buffer providing requests of switches the queue space to wait for been serviced.

Unlike most modes, the controller buffer here is finite with a buffer length *n*. An M/M/1/*n* queuing game model for a single switch single controller is formulated. Requests of switch ask the TCPL for services. After completing the service, the switch gets its benefit *R*. If the request joins the queue of the controller, it needs to pay the waiting cost *C* due to the existed *i* requests, for i=1,2,⋯,l, where *l* denotes queue lengths of the controller and $l\in N^{+}$. Whether a request joins the queue or not depends on the switch’s net benefit $S\left(i\right)$. Until now, the pure threshold strategy $l_{in}$ is an equilibrium strategy under individual optimization with the constraint condition $S\left(i\right)\geq 0$. However, under social optimization, the threshold $l^{*}$ is less than $l_{in}$ according to Section 3.2. Therefore, an admission fee is introduced to motivate requests to adopt the threshold $l^{*}$ rather than $l_{in}$.

### 3.2. Problem Description

In the system architecture, different users are connected to the network by the switch of the data layer and the data flow is realized through the information of the controller. The game players are the network nodes including the switch and the controller nodes in the system model and the strategies are the actions of the players in making decisions based on the optimal admission fees. The switch requests choose whether or not to join the queue by setting the equilibrium solution optimal admission fee of the queuing game. The competing resources are computing, storage, bandwidth and other network resources. In the model, they are embodied by the contention of network service resources for requests, that is, whether the switch node needs to get data transmission service and whether the controller node provides service. After the switch requests to accept the service of the controller, it will get benefits and pay the waiting cost when waiting in the queue. The controller gets its profit from the admission fee for requesting services. Obviously, different switch requests want the fastest service and the highest net revenue but the system resources are limited. Queuing game theory can exactly get the optimal strategy of the system and Nash equilibrium from the local and global point of view according to the switch’s benefit and the social welfare of the system. Therefore, the problem of 5G system resource allocation can be mathematically modeled by using queuing game theory and the optimal strategy of game participants can be obtained by solving the mathematical model, so as to obtain the optimal transmission control mechanism of the system. The relevant parameters are defined below.

**Definition** **1.**
*A stationary Poisson stream of the requests with parameterλ arrives in the TCPL. The service times of controller in SDN with parameterμ are independent, identically and exponentially distributed. For stability, it assumes that the system’s utilization factor $\rho =\frac{\lambda }{\mu }$ [19] satisfies ρ<1.*


**Definition** **2.**
*For a single switch single controller in TCPL, a switch pays an admission fee f to the controller for getting service. It obtains benefit R after being serviced completely. Once the request joins the queue, the switch costs C per unit of time to staying in the system. The switch decides to join the queue of the controller only if its net benefit $S\left(i\right)$ is non-negative.*


From the above definitions, the switch’s net benefit $S\left(i\right)\geq 0$ can be calculated according to the switch’s benefit, the admission fee and the waiting cost. If $S\left(i\right)\geq 0$, the request joins the queue. Then the controller’s profit $P\left(f\right)$ and social welfare $W\left(f\right)$ of the TCPL are obtained. In order to get optimal $P\left(f\right)$ and $W\left(f\right)$, the most suitable admission fee *f* has to been found. Specifications of the TCQG algorithm parameters are given in Table 1.

## 4. Transmission Control Scheme Using Queue Game

The software-defined transmission control mechanism will be designed according to the solution results of the system model. Firstly, the queuing game theory is applied to the simple model of a single switch single controller. Under the condition of maximizing the social welfare of the system, the optimal solution of the game is obtained. Then the conclusion is extended to the multi-switches single controller model to achieve an efficient system transmission control algorithm that does not rely entirely on the controller to collect global information.

### 4.1. Resource Optimization Model

The system resources required to request services included storage, computation, forwarding and other resources. The resource optimization model assumes that these resources are described by the controller providing the service through setting the waiting cost of the request in the queue, the admission fee and service rate. Whether switch requests join the queue of controller based on expected net revenue. This scheme can realize the optimal transmission control of the network through the establishment and solution of the model. The definition of parameters related to different requests can be extended according to the specific business of the network. The model defined in this section is the most basic network model.

In the single switch single controller model, the individual’s optimizing strategy of the switch is straightforward. A switch request who joins the queue expects a benefit $R-\frac{\left(i+1\right)C}{\mu }$ when *i* requests are already in the buffer and the one is served currently. If this value is non-negative, that is, if $i+1\leq \frac{R\mu }{C}$, the switch request enters. Otherwise, the request balks. As a result, an equilibrium strategy is obtained as $l_{in}$ with $l_{in}=\left\lfloor \frac{R\mu }{C}\right\rfloor$, where the $\left\lfloor \frac{R\mu }{C}\right\rfloor$ means the largest integer which is no more than $\frac{R\mu }{C}$.

For the same model, social optimization should be formulated to compare the result. The expected social welfare per unit of time is demoted by *W*. According to the queuing game theory,
(1)$$W=\lambda R\frac{1-\rho^{l}}{1-\rho^{l+1}}-C\left[\frac{\rho }{1-\rho }-\frac{\left(l+1\right)\rho^{l+1}}{1-\rho^{l+1}}\right].$$
where $\frac{1-\rho^{l}}{1-\rho^{l+1}}$ denotes the probability that a new request joins the queue and $\left[\frac{\rho }{1-\rho }-\frac{\left(l+1\right)\rho^{l+1}}{1-\rho^{l+1}}\right]$ denotes the expected number of requests in the buffer of the controller.

Through maximizing Equation (Equation 1), the queue threshold $l^{*}$ is calculated under social optimization. In order to confirm the relationship of size between $l^{*}$ and $l_{in}$, the effect of difference queuing models on the threshold strategy is discussed and analyzed as follows.

In Naor’s model [19], the FCFS discipline is assumed. When a new request joins the queue, it places the end. Therefore, the existed requests are not affected by the new arrivals. But for the future arrivals, this request joining the queue makes negative externalities. Because that the more requests join currently, the long waiting time for the future arrivals. The negative externalities are ignored which is the reason for a difference value between $l^{*}$ and $l_{in}$ under individual and social optimization respectively. In contrast, if we modify the queue model to LCFS-PR, each new request joins the queue and places the head. For the existed requests, the most negative externalities are imposed on and no more external effects for the future arrivals. What’s the difference is that the problem of whether a request joins the controller’s buffer or not turns to when the existed requests leave. Due to the irrelevant of the service’s order from the social view, the socially optimal queue threshold $l^{*}$ is the same under both FCFS and LCFS-PR disciplines.

Based on the above discussion, the model tries to determine $l^{*}$ as follows.

**Theorem** **1.**
*For a single switch single controller model of SDN, the buffer of the controller has to be set no less than the optimal queue threshold $l^{*}$ for maximizing the social welfare. The optimal queue length is constrained as*
(2)$$\left\{l_{m}\left|l_{m}=\left\lfloor l_{x}\right\rfloor ,\frac{R\mu }{C}=l_{x}+\frac{\left(1-\rho^{l_{x}-1}\right)\left(1-\rho^{l_{x}+1}\right)}{\rho^{l_{x}-1}{\left(1-\rho \right)}^{2}}\right.\right\}\leq l^{*}\leq l_{in}.$$


**Proof** **of** **Theorem** **1.**In LCFS-PR discipline, when *l* requests in front of a switch request, it reneges. *l* denotes the maximum queue length. Then the expected net benefit $s\left(l\right)$ is discussed for a request in position *l*,
(3)$$s\left(l\right)=Rp_{0}-C\bar{t},$$
where $p_{0}$ denotes the probability of the *l*th request getting service and $\bar{t}$ denotes the expected time in the buffer for this request.The *l*th request completes service and obtains the benefit *R* only if all the requests in front of it being serviced. Based on the queue theory, $p_{0}$ is calculated as
(4)$$p_{0}=\frac{1-\rho }{1-\rho^{l+1}}.$$Under the stable statement [30], the algebraic equations show as follows.For state 0, $\lambda_{0}p_{0}=\mu p_{1}$ such that $p_{1}=\frac{\lambda_{0}}{\mu }p_{0}$; For state 1, $\lambda_{1}p_{1}=\mu p_{2}$ such that $p_{2}=\frac{\lambda_{1}\lambda_{0}}{\mu^{2}}p_{0}$; …… For state l−1, $\lambda_{l-1}p_{l-1}=\mu p_{l}$ such that $p_{l}=\frac{\lambda_{l-1}\cdots \lambda_{1}\lambda_{0}}{\mu^{l}}p_{0}$.With the regularity condition $\sum_{j=0}^{l}p_{j}=1$, $\sum_{j=0}^{l}p_{j}=\sum_{j=0}^{l}\rho^{j}p_{0}=\frac{1-\rho^{l+1}}{1-\rho }p_{0}=1$
$\left(\rho \neq 1\right)$ can be got. Therefore, $p_{0}=\frac{1-\rho }{1-\rho^{l+1}}$.To get the expected time of the *l*th request, the LCFS-PR discipline can be redescribed as a gambler’s ruin problem [31]. The initial asset is *l*, the goal is l+1 and in each game round the winning probability is $p_{g}=\frac{\lambda }{\lambda +\mu }=\frac{\rho }{1+\rho }$, while the losing probability is $q_{g}=1-p_{g}=\frac{1}{1+\rho }$. As a result of the gambler’s ruin problem, the expected waiting time can be got as
(5)$$\bar{t}=\frac{1}{\lambda +\mu }\left[\frac{l}{q_{g}-p_{g}}-\frac{l+1}{q_{g}-p_{g}}\frac{1-{\left(\frac{q_{g}}{p_{g}}\right)}^{l}}{1-{\left(\frac{q_{g}}{p_{g}}\right)}^{l+1}}\right]=\frac{1}{\mu \left(1-\rho \right)}\left[l-\rho \left(l+1\right)\frac{1-\rho^{l}}{1-\rho^{l+1}}\right].$$Equation (Equation 3) can be rewritten as:
(6)$$s\left(l\right)=R\frac{1-\rho }{1-\rho^{l+1}}-\frac{C}{\mu \left(1-\rho \right)}\left[l-\rho \left(l+1\right)\frac{1-\rho^{l}}{1-\rho^{l+1}}\right].$$In the system model, the *l*th request can be completed service if $s\left(l\right)\geq 0$. Based on (6), the following Equation (Equation 7) can be got
(7)$$\frac{RC}{\mu }\geq \frac{l\left(1-\rho \right)-\rho \left(1-\rho^{l}\right)}{{\left(1-\rho \right)}^{2}}.$$The right-hand side changes as $\frac{l\left(1-\rho \right)-\rho \left(1-\rho^{l}\right)}{{\left(1-\rho \right)}^{2}}=\frac{1}{\left(1-\rho \right)}\left[l-\frac{\rho \left(1-\rho^{l}\right)}{\left(1-\rho \right)}\right]=\frac{l-\sum_{j=1}^{l}\rho^{j}}{\left(1-\rho \right)}=\sum_{j=1}^{l}j\rho^{l-j}=\sum_{j=0}^{l-1}\left(l-j\right)\rho^{j}$. Let $f\left(l\right)=\sum_{j=0}^{l-1}\left(l-j\right)\rho^{j}$. It is a strictly increasing function with *l*. According to (7), the maximum queue length $l^{*}$ exists with $f\left(l^{*}\right)\leq \frac{R\mu }{C}<f\left(l^{*}+1\right)$. Due to $f\left(l\right)-l=\frac{l\left(1-\rho \right)-\rho \left(1-\rho^{l}\right)}{{\left(1-\rho \right)}^{2}}-l=\frac{\rho }{1-\rho }\left(l-\sum_{j=1}^{l}\rho^{j-1}\right)\geq 0$, if $f\left(\alpha^{*}\right)=\frac{R\mu }{C}$, then $l^{*}=\left\lfloor \alpha^{*}\right\rfloor$ and $\left\lfloor \alpha^{*}\right\rfloor \leq \left\lfloor f\left(\alpha^{*}\right)\right\rfloor$. At last, the right side of in Equation (Equation 2) can be proved as
(8)$$l^{*}=\left\lfloor \alpha^{*}\right\rfloor \leq \left\lfloor f\left(\alpha^{*}\right)\right\rfloor =\left\lfloor \frac{R\mu }{C}\right\rfloor =l_{in},$$
which means that the socially optimal queue threshold $l^{*}$ is smaller than the individual optimal queue threshold $l_{in}$.In SDN, the service model generally is the FCFS discipline. In order to obtain the optimal social welfare, an admission fee *f* is introduced and paid to the controller by switch request who want to get service. Based on the model, the switch’s net benefit can be got as $R-\frac{\left(i+1\right)C}{\mu }-f$. If the net benefit is no less than zero, the request joins the queue of the controller. Then the maximal queue length is $\left\lfloor \frac{\left(R-f\right)\mu }{C}\right\rfloor$. Considering the social optimization condition, the maximal queue length can be deduced and the controller sets the corresponding fee as
(9)$$f=R-\frac{Cl}{\mu }.$$The controller’s profit is
(10)$$P\left(l\right)=\lambda \frac{1-\rho^{l}}{1-\rho^{l+1}}\left(R-\frac{Cl}{\mu }\right).$$The queue length satisfies the following conditions to maximize the profit:
(11)$$\left\{\begin{array}{c} P\left(l\right)>P\left(l-1\right)\\ P\left(l\right)\leq P\left(l+1\right) \end{array}\right..$$Substituting in (10), (11) can be rewritten as
(12)$$l+\frac{\left(1-\rho^{l-1}\right)\left(1-\rho^{l+1}\right)}{\rho^{l-1}{\left(1-\rho \right)}^{2}}\leq \frac{R\mu }{C}<l+1+\frac{\left(1-\rho^{l}\right)\left(1-\rho^{l+2}\right)}{\rho^{l-1}{\left(1-\rho \right)}^{2}}.$$Let $g\left(x\right)=x+\frac{\left(1-\rho^{x-1}\right)\left(1-\rho^{x+1}\right)}{\rho^{x-1}{\left(1-\rho \right)}^{2}}$, it is a monotone increasing function with *x*. The derivation of $g\left(x\right)$ is $g^{\prime }\left(x\right)=1+\frac{\ln\rho }{{\left(1-\rho \right)}^{2}}\left(\rho^{x-1}-\rho^{1-x}\right)$. Because 0<ρ<1, then $\frac{\rho^{x-1}}{\rho^{1-x}}=\rho^{2x}<1$, $\frac{\ln\rho }{{\left(1-\rho \right)}^{2}}<0$ and $g^{\prime }\left(x\right)>0$. Therefore, a unique solution $l_{x}$ to $g\left(x\right)=\frac{R\mu }{C}$ exists. Let $l_{m}=\left\lfloor l_{x}\right\rfloor$, then $l_{m}$ is the unique solution of the optimality conditions (12). The following equation is satisfied as
(13)$$g\left(l_{x}\right)=l_{x}+\frac{\left(1-\rho^{l_{x}-1}\right)\left(1-\rho^{l_{x}+1}\right)}{\rho^{l_{x}-1}{\left(1-\rho \right)}^{2}}=\frac{R\mu }{C},$$
where $\frac{\left(1-\rho^{l_{x}-1}\right)\left(1-\rho^{l_{x}+1}\right)}{\rho^{l_{x}-1}{\left(1-\rho \right)}^{2}}\geq 0$. It is immediate that $l_{x}\leq \frac{R\mu }{C}$ so that $l_{m}\leq \left\lfloor \frac{R\mu }{C}\right\rfloor =l_{in}$.To compare the values of $l_{m}$ and $l^{*}$, the relationship between the values of $f\left(x\right)$ and $g\left(x\right)$ is deduced firstly as $g\left(x\right)-f\left(x\right)=x+\frac{\left(1-\rho^{x-1}\right)\left(1-\rho^{x+1}\right)}{\rho^{x-1}{\left(1-\rho \right)}^{2}}-\frac{x\left(1-\rho \right)-\rho \left(1-\rho^{x}\right)}{{\left(1-\rho \right)}^{2}}=\frac{1-\rho^{x-1}-\left(x-1\right)\rho^{x}+\left(x-1\right)\rho^{x+1}}{\rho^{x-1}{\left(1-\rho \right)}^{2}}=\frac{\left(\sum_{j=0}^{x-2}\rho^{j}+\rho^{x}\right)-x\rho^{x}}{\rho^{x-1}\left(1-\rho \right)}$. With the constraint conditions 0<ρ<1 and x≥1, $g\left(x\right)-f\left(x\right)\geq 0$. Because $f\left(x\right)$ and $g\left(x\right)$ are monotone increasing functions, $\alpha^{*}\geq l_{x}$ is obtained so that
(14)$$l^{*}\geq l_{m}.$$Combining the in Equations (8) and (13), the optimal queue length can be got as (2).From what has been discussed above, the queue threshold $l_{m}$ is used in the model to achieve the optimal social welfare and the admission fee is set as
(15)$$f^{*}=R-\frac{Cl_{m}}{\mu }.$$  □

### 4.2. Optimal Transmission Control Algorithm Design

Extending the above conclusion to multi-switches single controller model of SDN, the TCQG algorithm is designed. Multi-switches mean the different arrival rates of requests, then it assumes that there exist *k* switches and $\lambda_{k}$
$\left(k\in N^{+}\right)$ denotes the corresponding arrival rate of switches. Therefore the controller’s profit is recalled as
(16)$$P\left(l\right)=\sum_{i=1}^{k}\lambda_{k}\frac{1-{\left(\frac{1}{\mu }\sum_{i=1}^{k}\lambda_{k}\right)}^{l}}{1-{\left(\frac{1}{\mu }\sum_{i=1}^{k}\lambda_{k}\right)}^{l+1}}\left(R-\frac{Cl}{\mu }\right).$$

**Theorem** **2.**
*For multi-switches single controller model of SDN, the competition of limited network resources arises among multiple switches. Based on the queuing game theory, the admission fee $f_{m}^{*}$ announced by the controller is*
(17)$$f_{m}^{*}=R-\frac{Cl_{m}^{*}}{\mu }\left\{l_{m}^{*}\left|l_{m}^{*}=\left\lfloor l_{x}^{*}\right\rfloor ,\frac{R\mu }{C}=l_{x}^{*}+\frac{\left[1-{\left(\frac{1}{\mu }\sum_{i=1}^{k}\lambda_{k}\right)}^{l_{x}^{*}-1}\right]\left[1-{\left(\frac{1}{\mu }\sum_{i=1}^{k}\lambda_{k}\right)}^{l_{x}^{*}+1}\right]}{{\left(\frac{1}{\mu }\sum_{i=1}^{k}\lambda_{k}\right)}^{l_{x}^{*}-1}{\left(1-\frac{1}{\mu }\sum_{i=1}^{k}\lambda_{k}\right)}^{2}}\right.\right\}.$$
*where $l_{m}^{*}$ and $l_{x}^{*}$ denotes the queue threshold and the unique solution to $g\left(x\right)=\frac{R\mu }{C}$ under the network conditions of multi-switches respectively.*


**Proof** **of** **Theorem** **2.**According to Burke’s theory [32], under a stationary state, the customer leave process of the queuing system M/M/1 is the Poisson process and the leave rate is equal to the arrival rate. In this model, the output of the switch is the input of the controller. All switches forward control information to the controller to which they belong in SDN. Then the arrival rate $\lambda_{c}$ of the controller is
(18)$$\lambda_{c}=\sum_{i=1}^{k}\lambda_{k}.$$Based on Theorem 1, recalling (13),
(19)$$g\left(l_{x}^{*}\right)=l_{x}^{*}+\frac{\left[1-{\left(\frac{1}{\mu }\sum_{i=1}^{k}\lambda_{k}\right)}^{l_{x}^{*}-1}\right]\left[1-{\left(\frac{1}{\mu }\sum_{i=1}^{k}\lambda_{k}\right)}^{l_{x}^{*}+1}\right]}{{\left(\frac{1}{\mu }\sum_{i=1}^{k}\lambda_{k}\right)}^{l_{x}^{*}-1}{\left(1-\frac{1}{\mu }\sum_{i=1}^{k}\lambda_{k}\right)}^{2}}=\frac{R\mu }{C}.$$Then the queue threshold $l_{m}^{*}$ of multi-switches single controller model is obtained as
(20)$$l_{m}^{*}=\left\lfloor l_{x}^{*}\right\rfloor .$$Based on (15), the admission fee $f_{m}^{*}$ can be calculated as
(21)$$f_{m}^{*}=R-\frac{Cl_{m}^{*}}{\mu }.$$In the system model, the control information of multi-switches is sent to the controller according to the transmission control strategy. A switch request enters to the TCPL and asks for service. The controller announces an admission fee and the current queue length to the switch. Then switch’s net benefit can be calculated. If the net benefit is no less than zero, the request joins the queue of the controller and waits for service. Therefore, the key to the TCQG algorithm is to determine the optimal admission fee. The TCQG algorithm flow chart is illustrated in Figure 2. By using the TCQG algorithm, the maximal social welfare of TCPL in a multi-switches single controller SDN model is obtained. The queuing game theory reduces the optimal strategy as an optimal admission fee. This parameter is easily added to flow tables so that the control flow information can achieve optimal transmission strategy.  □

## 5. Simulations and Comparisons

In this section, the performance simulation of the proposed TCQG algorithm in a TCPL of SDN is focused. Two simulations are provided to demonstrate the efficiency of the proposed model. In the first numerical simulation, the optimal admission fee is calculated under the SDN environment. By changing the value of system parameters, the performance of the scheme will be demonstrated. In the second system simulation, the end-to-end delay and the throughput of the contrast algorithm and the TCQG algorithm will be compared.

Firstly, the values of the common simulation parameters are defined in Table 2. In MATLAB 2018b, the theoretical value of the model is calculated and analyzed. By substituting these parameters in the Equations (2), (15) and (17), the optimal queue length and the admission fee of the system in the different models can be calculated. Then, with the optimal results, the TCPL obtains the maximum social welfare.

In order to simulate the influence of the congestion level to the network performance, the growing arrival rate is set as $\lambda \in \left(0,60\right)$ and the service rate is remaining unchanged. The system model is formulated based on the queuing game theory to get the optimal queue length by maximizing the controller’s profit and social welfare. The trend of the optimal result of the single switch single controller model is shown in Figure 3. The threshold queue length grows while the degree of congestion increases. The blue real curve represents the change of the optimal controller queue length with the growing before rounding down, while the green cross represents the value after rounding down. An equilibrium exists in the system. The more requesters there are, the more profit the controller will get. Then, the longer the queue will be, which in turn reduces the benefit for requests due to the increasing waiting cost. As a result, the optimal queue length of the controller is obtained.

According to the relationship of the threshold queue $l_{m}$ and the admission fee as the Equation (Equation 21), Figure 4 gives the optimal admission fee $f^{*}$ with the growing ρ. It observes that the admission fee decreases while more requests enter the queue. In order to maximize system welfare, the admission fee reduces for increasing the threshold queue. As a result, more requests are allowed to join the queue of the controller. Considering the negative effect of network congestion on performance, the model limits the increase of the growing waiting costs. At last, the optimal admission fee is set for transmission control.

The system’s welfare consists of the switch’s net benefit and the controller’s profit. As shown in Figure 5, the blue dash-dot line describes the controller’s profit and the red full line denotes the system welfare. The black star points out the queue length corresponding to the maximum welfare. When the queue length is less than $l_{m}$, the blue dash-dot line and the red full line almost coincide. In effect, the system’s welfare is more than the controller’s profit. After that, with the increase of the queue length, the system’s welfare declines faster than the controller’s profit. As a consequence, the optimal scheme is achieved by using the TCQG algorithm.

In the second realistic simulation, the average end-to-end delay and the throughput of the contrast algorithm and the TCQG algorithm are compared to show the improvement of the network performance. The Wardrop load-balancing algorithm of the literature [29] is selected to simulate the contrast transmission control scheme.

A realistic simulation environment is built by using the OMNeT++ 5.5.1. All the schemes and data sources are simulated on the Intel (R) Core (TM) i5-6200U CPU @ 2.3GHz using the C++ language. The basic network unit for SDN information transmission with the multi-switch single controller is structured. The relevant simulation Settings are assumed as follows. Switch requests are generated by the Poisson distribution with parameter λ. The arrival rate of controller requests is obtained with the Equation (Equation 18). The service rate of the controller follows the Poisson distribution with parameter ρ (λ<ρ) and the waiting cost of requests for the unit time is constant *C*. In the simulation process, the requests join to the controller based on whether the net benefit returned by the controller is non-negative or not. The net benefit is calculated using the conclusion of Theorem 2 about the admission fee. The specific parameters used are the same as the numerical simulation, as shown in Table 2. The TCQG algorithm runs for 100 s in each simulation experiment and the experiment is repeated 5 times with different random number seeds. The following simulation results are obtained.

As showed in Figure 6, the average end-to-end delay of the TCQG algorithm is lower on the whole. The reason is that the requests of the switch are serviced according to their net benefits. The waiting cost is also considered in the transmission control scheme. While in contrast algorithm, the primary goal of the scheme is the realization of traffic balance. The TCQG algorithm saves waiting time and then improves the transmission efficiency.

From the Figure 7, the bigger throughput is observed in our proposed algorithm. The explanation is that the designed TCQG algorithm utilizes the optimal admission fee as the system access parameters. It can improve the effectiveness of the system.

## 6. Conclusions

The SDN realize efficient data forwarding by separating the control function from the switch. It is widely concerned in 5G networks where massive amounts of data exist. This paper mainly solves the control information interaction scheme between switches and the controller and uses queuing game theory to carry out mathematical modeling of system units composed of single switch single controller. By solving the optimal solution of the system, the threshold queue length of the controller is calculated and then the optimal admission price, which is the key parameter of the system model, is obtained. The theoretical results are put into the system and the TCQG algorithm is designed. The switch calculates a net benefit of the current request based on the current queue length and admission fee returned by the controller. When the value is no less than zero, the request joins the queue of the controller and waits for service, otherwise, it reneges. The theoretical optimal solution ensures that the algorithm can maximize TCPL’s welfare. Simulation results show that the algorithm improves the operating efficiency of the system effectively.

The next work is to take into account the multi-stage series of queuing network and the output feedback at all levels. By modeling the system more in line with the real complex operation, the transmission control optimization scheme of the large responsible network will be obtained.

References

## Figures and Tables

**Figure 1 sensors-19-04170-f001:**
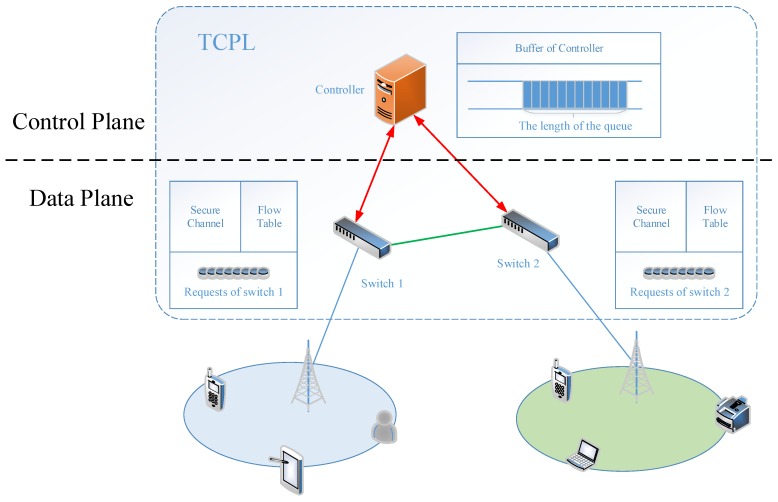
The transmission control system model using TCQG algorithm.

**Figure 2 sensors-19-04170-f002:**
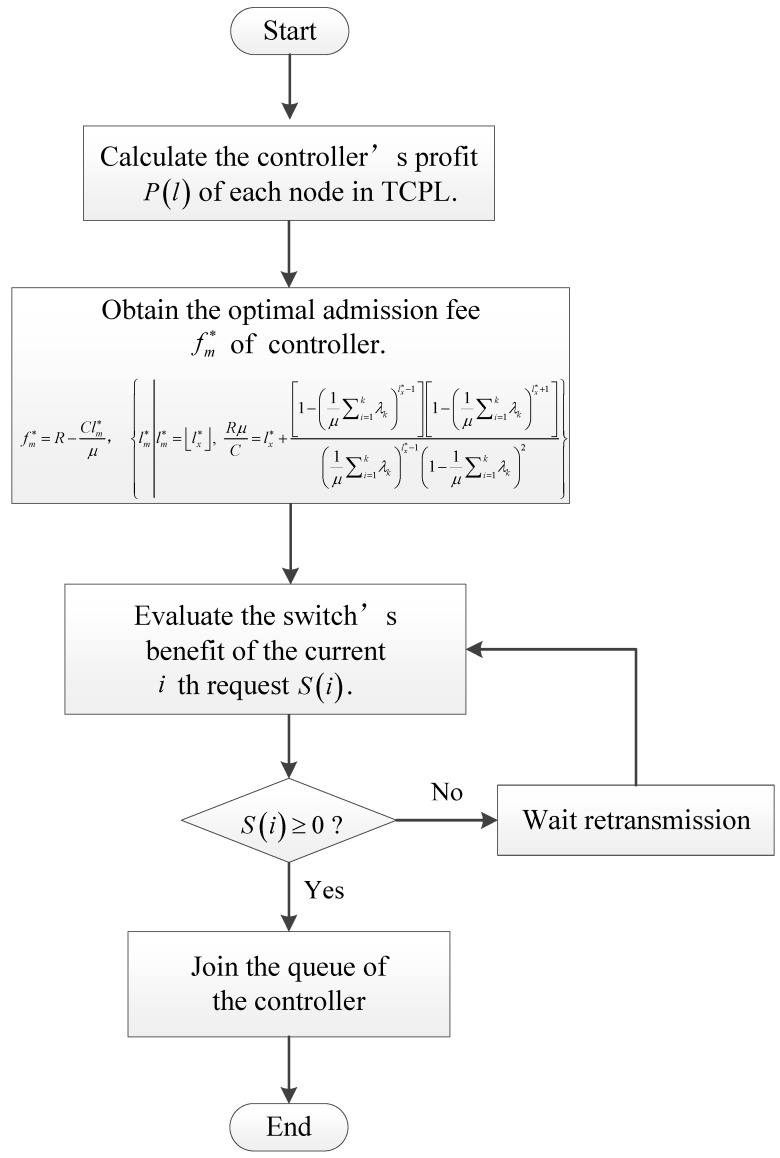
The flow chart of the TCQG algorithm.

**Figure 3 sensors-19-04170-f003:**
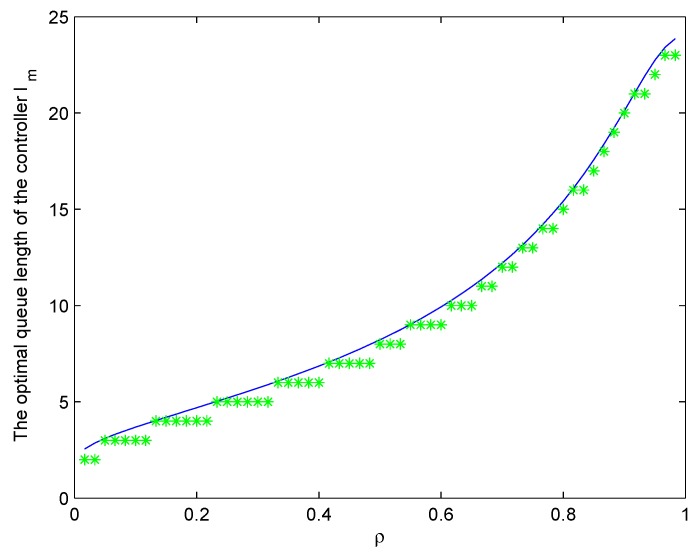
The optimal queue length of the controller for the different arrival rate.

**Figure 4 sensors-19-04170-f004:**
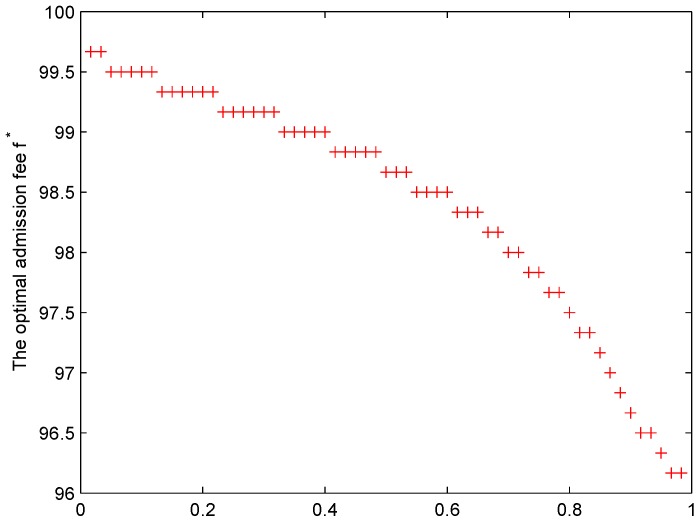
The optimal admission fee $f^{*}$ with the growing ρ.

**Figure 5 sensors-19-04170-f005:**
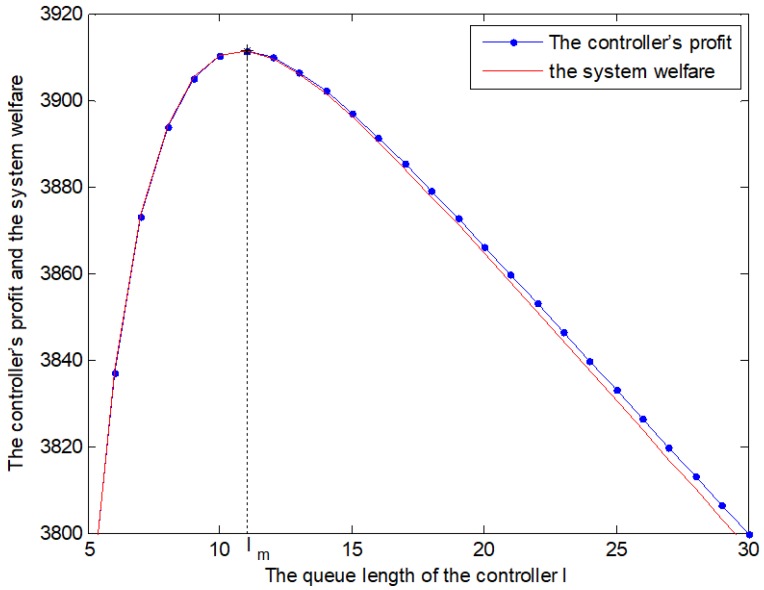
The controller’s profit and the system welfare with different queue length.

**Figure 6 sensors-19-04170-f006:**
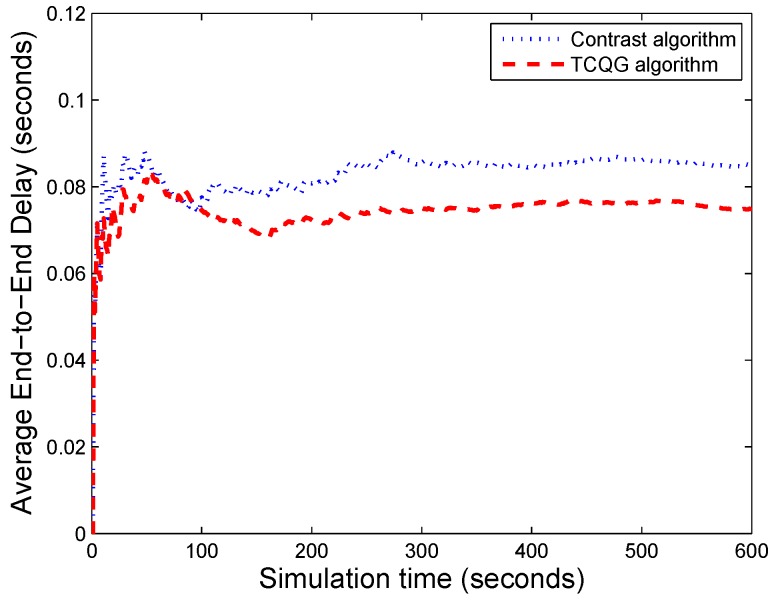
The comparison of average End-to-End Delay between two algorithms.

**Figure 7 sensors-19-04170-f007:**
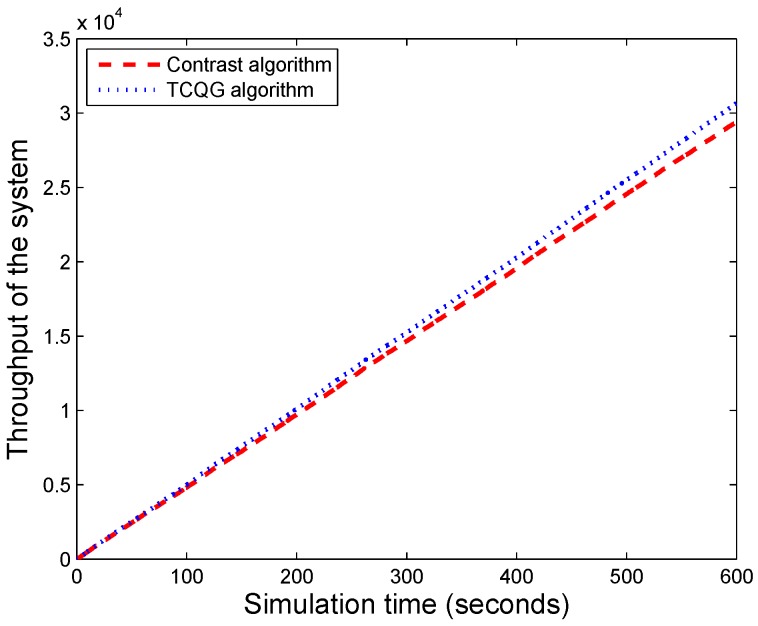
The comparison of throughput of the system between two algorithms.

**Table 1 sensors-19-04170-t001:** Specifications of the parameters of TCQG scheme.

Parameter Name	Meaning
λ	The arrival rate of request in TCPL
μ	Service rate of the controller in TCPL
ρ	Utilization factor $\rho =\frac{\lambda }{\mu }$
*l*	Queue lengths of the controller, $l\in N^{+}$
*f*	An admission fee *f*
*R*	A switch’s benefit from completing service
*C*	The cost of a switch to stay in the system per unit of time

**Table 2 sensors-19-04170-t002:** The value of the common simulation parameters.

Parameter Name	λ	μ	*C*	*R*
**Value**	(0,60)	60	10	100

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
