# Peer review of "TCQG—Software-Defined Transmission Control Scheme in 5G Networks from Queuing Game Perspective"

_sensors, 2019, doi:10.3390/s19194170_

Round 1

Reviewer 1 Report

This paper introduces an optimal software-defined 
transmission control algorithm based on queuing game theory over 5G network 

This sentence requires a proper reference: The primary goal of most applications is to be designed by minimizing 
power consumption, maximizing total network utilization, providing optimal load balancing, and 
other generic resource optimization techniques. 

It is not clear the benefits of using game theory for the proposal in the introduction.

It is not clear the main motivation about transmission control 
schemes. What are the benefits? What are the main drawbacks that need to be solved?

IMO, authors must introduce papers related to transmission control 
schemes. Specially, young references.

What are the players in this game? What are the competing resources to consider a game theory?

Figure 1 is difficult to read.

The paper must introduce the simulation methodology and parameters

Reviewer 2 Report

The paper uses classical queuing theory concepts to lay down a solution in the context of SDN switching requests. The interactions between switches and the controller are modeled with classical queuing theory concepts, and these are in turn the baseline of the TCQG algorithm presented in the paper. Most of the research in SDN is looking into practical approaches to implement networking solutions and new architectures. This paper complements the state of the art with proposals that can exploit well-know concepts (like queuing theory) in favour of more mature SDN-based solutions. 

In case the paper gets accepted, this reviewer strongly recommends a deep English proofreading.  

Reviewer 3 Report

In this paper, the authors investigate the control information interaction in software-defined 5G networks, and further propose to model the system using queue game theory. By solving the optimal solution, the authors obtains the key parameters of the system, i.e., the threshold queue length of the controller and the admission price. The topic is timely and interesting, and the whole paper is well structured. However, this paper should be improved by addressing the following issues before it can be accepted.

A proper and precise problem formulation is needed. The authors should implicitly describe what problem is going to be solved in this paper, to improve the network efficiency, or maximize maximize the revenue for the ISP? Please clarify who are the participants of the game and what resource they are compete for? Please clarify the meaning of request in this paper. It seems that the authors use two different kinks of requests in this paper: request of users and request of switch. What is the difference between them? The model is a bit confusing. I do not think this problem can be formulated without considering the capacity of substrate network. In other words, the controller cannot serve the request without considering the amount of resource each request is required. Please refer to recent literature on virtual network embedding (VNE). Moreover, the authors should revise this part thoroughly, make reasonable assumptions and formulate the problem with practical significance. It is better to reorganize Section 4.  This paper seems to find a optimal queue length and service fee for the controller. However, I cannot find the relationships with queue game and the equilibrium of it. The outcome of the queue game is just a reference. There are some typos and grammatical errors found, please polish the language thoroughly.

Reviewer 4 Report

Avoid using the term "we", write in 3rd person.

Chapter 5: It is not clear how the optimal queue length of the controller is obtained. Explain figures 3 and 4 better.

The validation of results would be significantly improved, with tests in experimental setup rather than simulated.

Round 2

Reviewer 1 Report

The authors improved the paper.

Follow some minor revision.

IMO, authors must introduce papers related to transmission control 
schemes. Specially, young references. I refer in the related work section.

The performance of TCQG must be compared with some related work.

Reviewer 3 Report

The authors have addressed all the concerns properly, the reviewer suggest to accept this paper as it is.

Author Response

Response to Reviewer 3 Comments:

Thank you very much for reviewing our paper. Thanks for your recognition of the content of our manuscript, which gives us great encouragement.

Yours sincerely,

Chao Guo

Reviewer 4 Report

I was clarified with the answers to my questions.

Author Response

Response to Reviewer 4 Comments:

Thank you very much for reviewing our paper. Thanks for your recognition of the content of our manuscript, which gives us great encouragement.

Yours sincerely,

Chao Guo